# SEMIPARAMETRIC REINFORCEMENT LEARNING

**Mika Sarkin Jain & Jack Lindsey** *
Stanford University
Stanford, CA 94309
{mjain4, jacklindsey}@stanford.edu

## ABSTRACT

We introduce a semiparametric approach to deep reinforcement learning inspired by complementary learning systems theory in cognitive neuroscience. Our approach allows a neural network to integrate nonparametric, episodic memory-based computations with parametric statistical learning in an end-to-end fashion. We give a deep Q network access to intermediate and final results of a differentiable approximation to k-nearest-neighbors performed on a dictionary of historic state-action embeddings. Our method displays the early-learning advantage associated with episodic memory-based algorithms while mitigating the asymptotic performance disadvantage suffered by such approaches. In several cases we find that our model learns even more quickly from few examples than pure kNN-based approaches. Analysis shows that our semiparametric algorithm relies heavily on the kNN output early on and less so as training progresses, which is consistent with complementary learning systems theory.

## 1 INTRODUCTION & RELATED WORK

We introduce a semiparametric approach to deep reinforcement learning, giving a neural network designed to predict Q-values access to the Q-values of previously observed state-action pairs and to their proximity to the current state-action pair according to a learned metric.

Parametric deep learning algorithms such as Deep Q Learning (Mnih et al., 2013) and its variants give state-of-the-art results on reinforcement learning tasks. Such algorithms use reward signals to update a large number of model parameters, typically used to estimate either a dynamics model or a Q function. An acknowledged drawback of many state-of-the-art reinforcement learning algorithms is their inability to learn without vastly more training experience than is needed by, say, a human (Tsividis et al., 2017). Nonparametric learning algorithms like k-nearest neighbors (kNN) offer a promising avenue to address this issue, as they support effective learning from few examples.

Recent work has incorporated a nonparametric algorithmic component into a model-free deep Q learning framework, providing an expressive alternative to parametric reinforcement learning. These algorithms maintain an episodic memory of low-dimensional embeddings of previously seen state-action pairs and estimate Q values based on stored values of proximal (according to a learned embedding metric) points in the memory. Blundell et al. (2016) use random projections onto a lower-dimensional space, while Pritzel et al. (2017) learn the embedding function with a neural network [1] Such models demonstrate substantial improvements in learning speed, and outperform competing algorithms on many tasks. However, their results suggest that fully parametric models are often more powerful asymptotically.

We introduce a "semiparametric" algorithm that combines parametric and nonparametric reinforcement learning approaches, aiming to capture the benefits of both. We employ an episodic memory module similar to Pritzel et al. (2017). However, we use the values and distance-based weightings of points in the episodic memory as *features*, which along with the current state-action embedding are used to compute a correction to the output of the kNN algorithm. This correction is determined by a neural network which can learn arbitrary functions of its inputs. Hence, we allow the network to

---

* Both authors contributed equally to this work.

[1] For ease of discussion, we will refer to a k-nearest neighbors-based as nonparametric even if the embedding function is learned using a parameterized model.

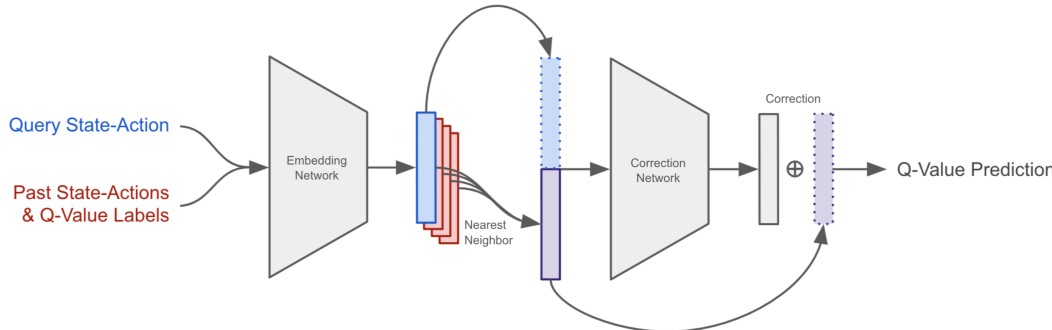

Figure 1: Architecture of semiparametric reinforcement learning for a single state-action pair. Gradients flow through the entire architecture.

overcome any deficiencies in performance arising from the use of kNN. Simultaneously, by granting the model access to the output of the kNN algorithm and the information contained in the episodic memory, we can obtain the learning benefits they provide.

Our algorithm is inspired by the theory of complementary learning systems in cognitive neuroscience (McClelland et al., 1995; Kumaran et al., 2016), which holds that a combination of episodic (in the hippocampus) and statistical (in the neocortex) learning is important for human task solving. The hippocampus can rapidly incorporate new information based on particular memories to define a rough working policy, while the neocortex learns more gradually but with greater power and generality. While prior work in episodic control focuses mainly on mirroring the hippocampal contribution to learning, our approach models the combination of these two systems.

## 2 METHODS

Our model consists of 1) a neural network that maps state-action pairs to an embedding space, 2) a fully differentiable kNN-based Q-value estimator that operates on a dictionary-based memory of past state-action pairs, each with a Q-value label and a representation in this embedding space, and 3) a second, "correction" neural network which predicts the Q-value of the current state-action, and operates on the embedding of the current state-action pair in addition to the distances and Q-values retrieved from memory by kNN.

This architecture is outlined in Figure 1. We trained our model on the Atari games, H.E.R.O., Bowling, Alien, and Enduro (Bellemare et al., 2013). We also trained it a simple Unity-based ball-rolling task (in which the objective is to collect twelve fixed tokens in a square field under a time constraint) in order to observe the algorithm's behavior all the way through convergence on a task given our resource constraints. Models details are outlined in Appendix B.

Our episodic memory module and differentiable approximation to k-nearest neighbors take after Pritzel et al. (2017). Throughout training, we store N-step approximations to Q-values of observed state-action pairs in a dictionary, along with their embeddings at observation time. We employ kNN to find a specified number (fixed at $K = 50$ in our experiments) of neighbors with embeddings $x_k$ and stored values $q_k$. The estimated kNN-based Q-value is computed as $\sum_{k=1}^{K} \left( \frac{q_k}{||x-x_k||^2} \right)^{-\tau}$.

The kNN-based Q-value estimate is modified by the output of a "correction network." We use a feedforward neural network acting on the concatenation of the current embedding and the nearest neighbors data $\{x_k, q_k\}$. Its output is added to the kNN-based estimate as a correction.

Our architecture choices were the result of extensive experimentation; see Appendix C for details.

## 3 RESULTS AND ANALYSIS

As our method was designed to capture the benefits of a nonparametric k-nearest neighbors-based approach as well as those of traditional powerful deep parametric models, we compare our model to

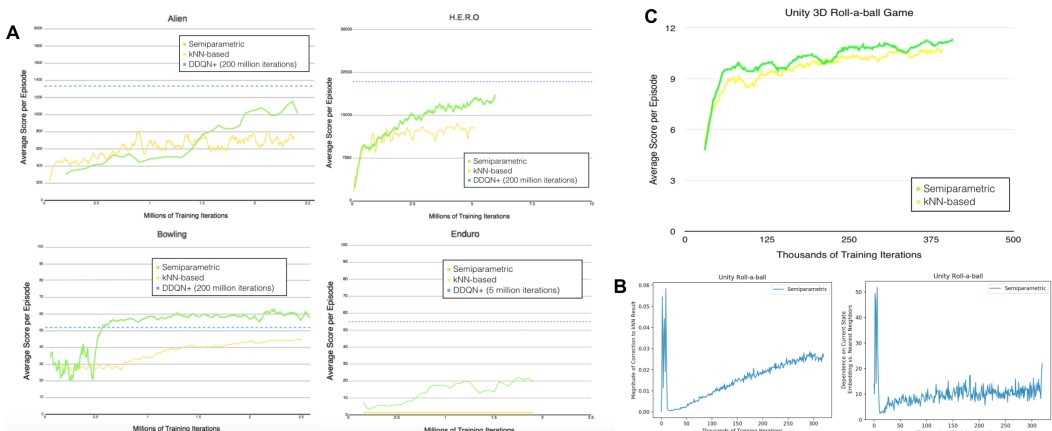

Figure 2: (A) Results on Atari games and (B) the Roll-a-ball task. (C) The relative dependence of Semiparametric learning on its nonparametric component decreases over training on Roll-a-ball.

high-performing models in either category as baselines–the Neural Episodic Control model (NEC) (Pritzel et al., 2017) and Double DQN with rank-based prioritized experience replay (which we will refer to as DDQN+) (Schaul et al., 2015). The set of tasks we experimented on includes one in which kNN-based learning proceeds most quickly and exceeds all other models (Bowling, Alien), one in which it fails to learn while DDQN+ gives good performance (Enduro), and one in which kNN-based learning exceeds DDQN in early training stages but lags in final performance (H.E.R.O.). Resource limitations prevented us from training most models to convergence; for baseline models, we use dashed lines to indicate asymptotic or intermediate performance figures taken from published results.

In early learning stages on the models tested, semiparametric learning matched or exceed NEC, which itself learns much faster than DDQN+ on all games but Enduro (Pritzel et al., 2017; Schaul et al., 2015). On H.E.R.O., semiparametric learning matched NEC in early learning stages (Figure 2A). On Bowling, it gave even faster early learning. We conclude that our method captures and sometimes enhances the early-learning advantage of nonparametric methods like NEC. This result is reasonable, as the decision in the NEC to weight neighbors according to inverse distance in embedding space from the current state-action is rather arbitrary. A more complex function of these distances, or one dependent on the current state embedding, might more accurately estimate the Q-function. Our method provides exactly this functionality.

Time and resource restrictions prevented us from obtaining asymptotic results on all games. However, in all but one settings tested, semiparametric learning exceeded the lower-performing of the two baselines. H.E.R.O. is representative of a substantial number of Atari games in which NEC learns more quickly than DDQN+ but has inferior final performance. We found that semiparametric learning exceeds the asymptotic performance of NEC (Figure 2A). This result extended to games in which nonparametric learning gave poor results even in early stages. On Enduro, where NEC fails to learn, semiparametric learning did not suffer the same issue. The semiparametric model exhibited the same advantage on the Roll-a-ball task, where we could train to convergence (Figure 2B).

We investigated our model's dependence on nearest-neighbors data relative parametric learning over the course of learning. We quantified this dependency with two approaches: 1) calculating the magnitude of the correction to the k-nearest neighbors output over the course of training, and 2) by measuring the ratio of magnitudes of the gradient of the Q-value estimates with respect to the current state embedding and with respect to the episodic memory data. In the most tasks, including the Roll-a-ball task (Figure 2B), semiparametric learning appears to rely increasingly on its parametric corrections to the kNN results. This is consistent with the paradigm that purely nonparametric, kNN-based methods become less advantageous as training progresses.

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

## 4 APPENDIX A: ALGORITHM

---

**Algorithm 1** Semiparametric Reinforcement Learning

---

1: $\mathcal{D}$: replay memory.
2: $M_a$: a dictionary for each action $a$.
3: $N$: horizon for $N$-step $Q$ estimate.
4: **for** each episode **do**
5:    **for** $t = 1, 2, \ldots, T$ **do**
6:       Receive observation $s_t$ from environment with embedding $x$.
7:       Compute $x_k, q_k$ of k-nearest neighbors to $x$
8:       Compute $Q_{kNN}(s_t, a)$ for each action $a$ via $P_{kNN}(c) = \sum_{k=1}^{K} \left( \frac{c_k}{||x-x_k||^2} \right)^{-\tau}$ from $M_a$
9:       Compute $\delta Q$ via correction network with inputs $x, x_k, q_k$
10:      Estimate $Q(s_t, a) \leftarrow Q_{kNN}(s_t, a) + \delta Q$
11:      $a_t \leftarrow \epsilon$-greedy policy based on $Q(s_t, a)$
12:      Take action $a_t$, receive reward $r_{t+1}$
13:      Append $(x, Q^{(N)}(s_t, a_t))$ to $M_{a_t}$.
14:      Append $(s_t, a_t, Q^{(N)}(s_t, a_t))$ to $\mathcal{D}$.
15:      Train on a random minibatch from $\mathcal{D}$.
16:    **end for**
17: **end for**

---

## 5 APPENDIX B: METHOD DETAILS

Our model is trained using RMSProp (Tieleman and Hinton, 2012) with a learning rate of 0.00001, decay of 0.9, and $\epsilon = 0.01$. We use experience replay with replay memory size 100000 and update Q-values using batches of size 32. We store episodic memories in a dictionary of size 500000 corresponding to the most recently observed state-action pairs, computing values based on $N$-step Q-value estimates with $N = 100$. We employed a reward discount factor of 0.99. We performed an approximation to k-nearest neighbors using the Python Annoy library and $K = 50$. The total number of training iterations varied due to time and resource limitations.

On the "roll-a-ball" task, our initial network maps to an embedding space of size 128. It consists of a feedforward network with two permutation-invariant layers (Guttenberg et al., 2016) followed by a fully connected layer. On the Atari games, our initial network maps to an embedding space of dimension 128 and consists of a convolutional neural network identical to the first three layers of the model described in Mnih et al. (2013), followed by a fully connected layer. In all cases we obtain a 128-dimensional embedding $x(s, a)$ of the current state-action pair. Hidden layers of the correction network (three layers) had size 32 for the Unity roll-a-ball task and 256 for Atari games.f

Though this algorithm is technically non-differentiable at points in parameter space where the identities of the k-nearest neighbors chosen change, these cases form a set of measure zero in parameter space and hence may be ignored when performing backpropagation. Hence, we may treat our model as fully differentiable. This allows the initial embedding function to be learned to maximize performance of the later stages of the algorithm. We omit the updates to existing dictionary values employed in Pritzel et al. (2017), as these slow down the algorithm considerably.

## 6 APPENDIX C: ATTEMPTED ALTERNATIVE MODELS

We experimented with a number of alternative semiparametric architectures. In one, the kNN distances and Q-values were used only as input to the correction network, and thus what we have referred to as the "correction network" in fact computed Q-value estimates itself. This architecture proved unsuccessful on most domains, though its hyperparameter space was not fully explored. In another, we appended the Q-values and distances retrieved by kNN to each layer of the correction network rather than to just the first. This model had slightly worse performance to the one we have described. We also experimented with weight regularization and regularizing the magnitude of the

kNN correction as to incentivize the our model to adhere more closely to the kNN results. These regularization methods did not result in discernible performance benefits on the tasks studied.

