# OpenReview forum: "Semiparametric Reinforcement Learning"
_ICLR.cc/2018/Workshop — Accept_

### Official Review · AnonReviewer2 · 2018-02-25
**Straightforward extension to NEC to include a parametric correct**

**Rating:** 7
**Confidence:** 4

**Review:**

NEC (Pritzel, 2017) demonstrated a non-parametric approach to scalable, model-free RL, by using a learned, neural network for embedding states in a KNN and querying this to estimate state-action values. NEC was significantly more data efficient than most parametric approaches, but in many environments performance plateaued below that of parametric model-free

This work extends NEC by, in addition to estimating state-action values as in NEC, learning a parametric corrector network which estimates a correction based on the query state embedding, all retrieved past-state embeddings and values. This is demonstrated in several environments to outperform NEC.

Pros:
- Fairly well-communicated result.
- Topic of significant interest.
- Good baseline comparisons.

Cons:
- A relatively straightforward extension to NEC,
- Relatively modest empirical improvements (compared to prior work).

Minor:
Tau (appears the Q-value estimates) is never explained. Is the normalization here done differently than in NEC?

---

### Official Review · AnonReviewer1 · 2018-03-09
**a good paper**

**Rating:** 7
**Confidence:** 2

**Review:**

This paper proposed a new learning scheme --- semiparametric reinforcement learning. This new approach lets a neural network to integrate nonparametric, episodic memory-based computations with parametric statistical learning in a seamless fashion. The method also has several advantages --- early-learning (associated with episodic memory-based algorithms) and reduced asymptotic performance disadvantage.

Overall the paper is well-written. The methodology within the paper appears to be reasonable and very interesting to me. There is enough empirical results as well as technical details for a workshop paper. Because this is not my research area, I cannot judge its technical contribution.

---

### Official Review · AnonReviewer3 · 2018-03-11
**Insufficient theoretical and experimental evidence for validity**

**Rating:** 3
**Confidence:** 4

**Review:**

Summary:
The Authors propose to extend the NEC algorithm by adding an additive correction term to its output. The correction term is computed by a Neural Network that takes as input the concatenation of the current state-action embedding as well as the embeddings and stored values of all 50 nearest neighbours that are used in the NEC algorithm.

Novelty (6/10):
While building on an existing algorithm, it proposes a non-trivial extension to it.

Clarity (4/10):
Points subtracted because I couldn't find any explanation or citation for the "Roll-a-ball" game.
Also, several important hyperparameters/explanations are missing, in particular:
- What do "Training Iterations" mean in the performance plots?
- If it does mean gradient updates, how many frames are seen between each gradient update?
This information is important as one goal of the algorithm is to improve learning speed compared to standard parametric methods.

Significance (8/10):
I believe the question of how to combine parametric and nonparametric solutions is highly relevant for Deep RL as it has the potential of overcoming the problem of large required sample sizes before acceptable results are achieved, a major problem in many (real world) domains.

Quality (3/10):
I don't believe the suggested idea in it's current state is far enough developed.
I neither find the experimental evidence nor the idea by itself convincing enough, as explained below.

Regarding the idea:
I don't find the concatenation of all 50 nearest neighbours and their values to compute the correction term (which is the main contribution of the paper) an obviously reasonable solution. The experimental evidence is too weak to convince me otherwise.
- It creates an extremely large fully connected layer to combine all that information: About 1.7M parameters just in one layer for Atari despite only a 128 sized hidden layer (256 for each state embedding). Admittedly that is in the same order of magnitude than the entire DQN-network so it's not terrible, but it's certainly not an inventive or elegant solution.
- The fully connected layer does not take into account the permutation invariance of the set of those 50 nearest neighbours. They don't mention any ordering of the set (e.g. by distance), so I assume the order is random and thereby permutation invariant.

I'm also not sure whether the fundamental idea of the correction architecture makes sense:
Since the idea of having a tabular representation in NEC is that this will adapt faster to new information than a NN, the target for the NN (the correction term) is changing faster than the NN is likely to be able to adapt.
In later stages of training when the parametric model starts to outperform the nonparametric one, the target for the correction-NN is potentially more difficult to fit to than in the standard parametric case, thereby hurting performance: Now, instead of targeting the state-action value directly, it targets the deviation of the kNN algorithm (potentially still fast changing) from the state action value.
I might be missing something, but a discussion of this issue should be included in the paper as I don't believe it to be obvious of why it is supposed to work.

Consequently, their claim that their architecture improves performances over NEC in early phases of training is therefore interesting but not backed up enough.

Experimental evidence:
- More and longer runs are needed. A single run is not representative in Atari as the variance can be quite large.
- The information is not given in the paper, but if "Training Iterations" (x-axis on plots) means gradient updates and they also collect 16 frames in between updates (like in Pritzel et al), then they do not convincingly outperform NEC, as 2.5M updates correspond to 40M frames.
  - Alien: ~1100 (Semip.) vs ~4500 (NEC)
  - HERO: ~18000 (Semip., after 80M frames) and ~15000 (Semip. after 40M frames) vs ~17000 (NEC, 40M frames)
  - Bowling: ~60 (Semip.) vs ~80 (ENC)
  - Enduro: ~20 (Semip., after ~32M frames) vs ~0 (NEC)
(All results are estimated by looking at the performance plots, the value at around 40M frames is reported unless stated otherwise)


In conclusion:
Pros:
- Very relevant and significant topic

Cons:
- Key information missing from paper (Roll-a-ball & how many frames are seen)
- Insufficient experimental results (Not convincingly outperforming any baseline)
- Insufficient theoretical analysis/discussion of their method

---

### Decision · Program_Chairs · 2018-03-20
**ICLR 2018 Workshop Acceptance Decision**

**Decision:**

Accept

**Comment:**

Congratulations, your paper was accepted to the ICLR workshop.